

# Relative dispersion in the South Western Mediterranean as derived from satellite-tracked surface drifting buoys

Maher Bouzaiene[1], Milena Menna[2], Pierre-Marie Poulain[2], Dalila Elhmaidi[1]

[1]Université de Tunis El Manar, Unité de recherche: Rayonnement thermique, Faculté des Sciences de Tunis, Tunisie

5  [2]National Institute of Oceanography and Experimental Geophysics, Sgonico (Ts), Italy

*Correspondence to*: mmenna@inogs.it

**Abstract.** Relative dispersion ($D^2$) in the South Western Mediterranean is analyzed using surface drifter pairs deployed during the period from 1986 to 2016. The results show the existence of four well-known regimes. The first regime, characterized by an exponential increment of the relative dispersion (Lundgren or exponential regime), corresponds to the chaotic advection at small scales and small separation distances, lasts for a few days. In the second regime, extending from 1.5 to roughly 7 days, for scales between 25 and 57 km and 1-3 km of initial distance, $D^2$ increases as time cubed (Richardson regime). The third regime occurs for initial distances of 5-10 km and times of 1.5-13 days; $D^2$ increases quadratically with time (Ballistic regime). The forth regime corresponds to time scales larger than 34 days for initial distances of 1-3 km and to 23 days for 35-40 km with a linear increase in time of $D^2$ (Rayleigh or diffusive regime). The relative diffusivity and characteristic dispersion time exhibit three different phases based on the initial pair separations and corresponding with Lundgren, Richardson and Rayleigh regimes, respectively. In the first phase (enstrophy cascade range) the diffusivity is $\sim D^2$ for distances smaller than 15 km and initial separation distances between 5 km and 10 km, and also for distances smaller than 40 km for initial separation distances between 35 km and 40 km; characteristic dispersion time is constant. In the second phase (inverse energy cascade), the diffusivity and characteristic dispersion time increase with growing distances following the 4/3 and 2/3 power laws, respectively, for scale ranging between 3 and 15 km and for initial distances smaller than 3 km. The third phase occurs for distance larger than 55 km, all pair velocities are uncorrelated and both relative diffusivity and characteristic dispersion time are approximately constants.

## 1 Introduction

The surface circulation in the Western Mediterranean is relatively complex due to the presence of coherent vortices that have
25  a great impact on the distribution of water masses and the evolution of biological quantities. The mean basin-scale circulation is cyclonic (Poulain et al., 2013) and the zonal motion is dominant (Nefzi et al., 2014), in particular in the Algerian sub-basin. Currents move from west to east along the Algerian coast, with velocities as large as 30 cm/s (Poulain et al., 2012), and develop intense eddies with different spatial scales and a maximum diameter of about 200 km (Testor et al., 2005). These eddies, generated by the instability of the along-slope current (Millot, 1999), affect strongly the local ecosystem and induce a
30  crucial variability of marine species (Taupier-Letage et al., 2003). The South Western Mediterranean (SWM; Figure 1) is characterized by the inflow of Atlantic Water (AW) from the Atlantic Ocean through the Strait of Gibraltar, therefore the study of particle dispersion in this region can add new information about the transport of AW and its mixing with Mediterranean waters.



The relative dispersion, which is the measure of the mean square separation distance of particle pairs, provides a description of the spreading of fluid particles under chaotic advection and turbulent motions (Schroeder et al., 2011). It is used in many practical applications, such as for understanding and predicting the spreading of pollutants and biological quantities in the ocean (Koszalka et al., 2009; Schroeder et al., 2011), and is defined as follow (Richardson, 1926; Babiano et al., 1990;

Ohlmann et al., 2012):

$$D^2(t,D_0) = \left\langle (X_i(t,D_0) - X_j(t,D_0))^2 \right\rangle \tag{1}$$

where $\langle . \rangle = \dfrac{1}{N_P} \sum_{i \neq j} (.)$ is a Lagrangian average on all pairs, $N_P$ is a number of drifter pairs, indices $i$ and $j$ are for particles,

$X_i(t,D_0)$ and $X_j(t,D_0)$ are the vector positions of the particle displacements from their initial positions $D_0$.

In case of two-dimensional, stationary turbulence the relative diffusivity and the characteristic dispersion time are defined

respectively by (Babiano et al., 1990):

$$Y(t,D_0) = \frac{1}{2} \frac{d\left(D^2(t,D_0)\right)}{dt} \tag{2}$$

$$\tau(t,D_0) = \frac{D^2(t,D_0)}{Y(t,D_0)} \tag{3}$$

The relative diffusivity is related to the mean square relative velocity $\left\langle \delta v^2(t,D_0) \right\rangle$, used in homogeneous, stationary, and isotropic two-dimensional turbulence to analyze the correlated or uncorrelated pair velocities (Babiano et al., 1990; Koszalka

et al., 2009; Ohlmann et al., 2012):

$$\begin{aligned}
<\delta v^2(t,D_0)> &= <(u'_i(t,D_0) - u'_j(t,D_0))^2> \\
&= 2<u'^2_i> -2<u'_i u'_j> \\
&= 4E - 2<u'_i u'_j>
\end{aligned} \tag{4}$$

Where $u'_i$ and $v'_j$ are the residual velocities of particles (i.e. $u'_i(t) = u_i(t) - \langle u_i(t) \rangle$) considered in pairs and $E$ is the eddy

kinetic energy defined as $E = \frac{1}{2} \left\langle u'^2_i \right\rangle$. If the particle velocities become uncorrelated, the mean square relative velocity

tends to a value of four times the eddy kinetic energy.

In the last decades several studies have been focused on the relative dispersion in many parts of the World Ocean and Mediterranean Sea using in-situ measurements. These studies identified in total four regimes that depend to the time $t$, the separation distance between the pairs $D$, and to the forcing injection scale $D_I$. In two-dimensional turbulence, $D_I$ is the scale at which energy is injected to coherent vortices and is estimated as the internal Rossby radius of deformation. $D_I$ is of about 30 km in the ocean (Ollitraut et al., 2005) and it is of 10-15 km in the Mediterranean Sea (Robinson et al., 2001). For

$D_0 >> D_I$ the particle are far from each other and the pair velocities become uncorrelated (Babiano et al., 1990; Ollitraut et al., 2005; Sansón et al., 2017). The above mentioned four regimes are:

- the Lundgren or exponential regime that occurs at early time and small separation scale, the relative dispersion grows exponentially in time (Lundgren, 1981; Bennett, 2006; LaCasce, 2010). The dispersion is non local and advected by eddies with scales larger than the separation distance. Under the exponential growth, the relative

diffusivity should scale as $Y(t,D)_0 \sim D^2$ in the enstrophy cascade range (Lin, 1972) for scales smaller than $D_I$.





- the Richardson regime that takes place at intermediate times for small initial distances; this local regime is characterized by pair spreading driven by eddies that are at the same scale as the separation distance, and the relative dispersion grows as time cubed $D^2(t,D_0) \sim t^3$ (Bennet, 1984; Babiano et al., 1990). Under the Richardson law, the mean square relative velocity is $<\delta v^2(t,D_0)> \propto t$ (Babiano et al., 1990; Ollitrault et al., 2005), the relative diffusivity should scale as $Y(t,D_0) \sim D^{4/3}$ (Richardson, 1926) and the characteristic dispersion time should scale as $\tau(t,D_0) \sim D^{2/3}$, in which energy is transferred to large scale in the inverse energy cascade range (Babiano et al., 1990).

- the Ballistic regime that is characterised by intermediate times and intermediate initial distances; relative dispersion increase quadratically $D^2(t,D_0) \sim t^2$.

- the Rayleigh or diffusive regime that starts at sufficiently long time, large initial distance ($D_0 \gg D_l$) and large pair separation. The relative dispersion exhibits a linear growth in time ($D^2(t,D_0) \sim t$ local regime) and the mean square relative velocity becomes just four times the eddy kinetic energy $\langle \delta v^2(t,D)_0 \rangle \propto 4E$. In this case the pair velocities become uncorrelated and the relative diffusivity becomes constant at large scale and equal to twice the absolute diffusivity (LaCasce and Bower, 2000; Ollitraut et al., 2005).

The exponential regime was found: in the Gulf of Mexico for the first 2-3 days with $D_0 < 1$ km and separation distance between 40 and 50 km (La Casce and Ohlmann, 2003), in the Santa Barbara Channel for the first 5 hours of sampling with $5 < D_0 < 10$ m (Ohlmann et al., 2012), in the Gulf of California for the first few days with $D_0 < 10$ km (Sansón, 2015) and in the southwestern Gulf of Mexico for the first 3 days with $D_0 < 2$ km (Sansón et al., 2017). In the Mediterranean Sea the relative dispersion increases exponentially with time for the first 6 hours in the Gulf of La Spezia (Haza et al. 2010), for time scale between 4 and 7 days in the Liguro-Provençal basin with $D_0 < 1$ km and separation distance of 10-20 km (Schroeder et al., 2011), for the first day in the Adriatic Sea with $D_0 < 1$ km (Poulain et al., 2013) and for the first 3 days in the Western Mediterranean with $D_0 \sim 5$ km (Nefzi et al., 2014).

The Richardson regime was observed in the Nordic Seas for time scale between 2 and 10 days and $D_0 < 2$ km (Koszalka et al., 2009), in the Gulf Stream for $D_0 < 700$ m and separation distance of 1-3 km (Lumpkin and Elipot, 2010), in the Gulf of California for $D_0 < 10$ km and separation distance larger than 3 km, in the southwestern Gulf of Mexico for $D_0 < 2$ km and separation distance between 10 and 150 km (Sansón et al., 2017) and in the Western Mediterranean for $D_0 \sim 5$ km and time scale of 3-20 days.

The Ballistic regime follows the exponential phase in the Santa Barbara Channel for time scale larger than 5 hours (Ohlmann et al., 2012) and in the Adriatic Sea for time scale of 2-10 days and separation distance between 10 and 30 km (Poulain et al., 2013).

The diffusive regime was found in the Gulf Stream for separation distance of 300-500 km (Lumpkin and Elipot, 2010), in the Adriatic Sea for time scale larger than 10 days and separation distance larger than 30 km (Poulain et al., 2013), and in the Western Mediterranean for $D_0 \sim 5$ km and time scale larger than 20 days (Nefzi et al., 2014).

In the present work, we use in-situ drifter data to study the surface relative dispersion in the SWM, a region that plays a key role on the dispersion and the transport of surface water masses from the Atlantic to the Eastern Mediterranean Basin. Our goal is to study, describe and understand the horizontal transport of mixing proprieties in a two dimensional flow. In-situ data and drifter pairs are presented in section 2. The results of the statistical properties of the relative dispersion, the relative



diffusivity, the mean square relative velocity and the characteristic dispersion time are discussed in section 3, and conclusions are presented in section 4.

## 2 Data and methods

The data set analyzed for this study derives from a total of 213 drifters deployed in the SWM during the period 1986-2016. These drifters are mainly of two types: Coastal Ocean Dynamics Experiment (CODE) drifters designed to measure the currents in the first meter near the surface layer (Davis, 1985; Poulain, 1999), and Surface Velocity Program (SVP) drifters, equipped with a holey sock drogue centred at 15 m depth, a sea surface temperature sensor and a tension sensor that allow checking the presence of the drogue (Sybrandy and Niiler, 1991; Lumpkin and Pazos, 2007). Drifter Argos and GPS positions were tracked by the Argos Data Collection and Location System (DCLS - carried by the NOAA polar-orbiting satellites) or via the Iridium satellite system. Drifter position time series were first edited from spike and outliers, then linearly interpolated at regular 6-h intervals using the kriging technique (Hansen and Poulain, 1996). The interpolated positions were low-pass filtered using a Hamming filter (cut-off period at 36 h) in order to remove high frequency current components (tidal and inertial currents) and were finally sub-sampled at 6-h intervals. Velocity components were then estimated from centred finite differences of 6-h sub-sampled positions (Menna et al., 2012; Poulain et al., 2012; Poulain et al., 2013). Among all the drifters deployed in the SWM we have selected for this study: 18 deployment pairs in 1-3 km, 50 deployment pairs in 5-10 km, 16 deployment pairs in 35-40 km and 9 deployment pairs in 55-60 km (see Figure 1). The deployment drifter pairs were identified as those deployed together (original pairs); drifter pairs not launched together (independent chance pairs), were also considered (Morel and Larcheveque, 1974; Er-el and Peskin, 1981; LaCasce and Ohmann, 2003). When an independent chance pair is found, our procedure fixes the pair time and wait 5 days after this time before to search for another independent pair. In this way we found: 74 chance pairs in 1-3 km, 165 chance pairs in 5-10 km, 283 chance pairs in 35-40 km and 337 chance pairs in 55-60 km (see Figure 1). To improve the accuracy of our results, we considered both the original and chance pairs following LaCasce et al., 2000. The number of drifter pairs increased with distance: 92 pairs deployed in 1-3 km, 215 pairs deployed in 5-10 km, 299 pairs deployed in 35-40 km and 346 pairs deployed in 55-60 km. The number of pairs decreased with time after deployment, due to the finite operating life of the drifters (see inserts in Figure 1).

## 3 Results

The temporal evolution of the mean square relative velocity, $\left\langle \delta v^2(t, D_0) \right\rangle$, and four time kinetic energy, $4E$, were used to describe the behavior of pairs and estimate the time at which the drifter pair velocities become uncorrelated (Figure 2). Largest values of the kinetic energy were found in the first days, because drifters were generally deployed in energetic features like fronts. The temporal evolution of $4E$ shows larger energy values with respect to $\left\langle \delta v^2(t, D_0) \right\rangle$ (Figures 2a, 2b and 2c), except when the initial pair separations is of 55-60 km (Figure 2d); in this condition the curves are reversed and $\left\langle \delta v^2(t, D_0) \right\rangle$ shows more energetic values. Larger values of $4E$ with respect to $\left\langle \delta v^2(t, D_0) \right\rangle$ mean that $2\left\langle u_i' u_j' \right\rangle > 0$ (see Eq. 4), hence the zonal and/or meridional components of the pair velocities have to be of the same sign (positive or negative), therefore they have the same directions; in this situation the pair velocity components are correlated. In contrast, larger values of $\left\langle \delta v^2(t, D_0) \right\rangle$ with respect to $4E$ mean that $2\left\langle u_i' u_j' \right\rangle < 0$, hence the pairs of zonal and/or meridional components have



opposite signs. This condition occurs for the majority of pairs with initial separation distances between 55 km and 60 km, when these pairs move in the main mesoscale eddies of the SWM (see Figure 3), and leads the velocity components to be anticorrelated.

When $4E$ and $\left\langle \delta v^2(t, D_0) \right\rangle$ reached about the same values (red lines in Figures 2a, 2b and 2c), the pair velocities become

uncorrelated ($2\left\langle u_i' u_j' \right\rangle = 0$). Identical values are detected at ~ 34 days for pairs deployed in 1-3 km, at about 30 days for pairs deployed in 5-10 km and at about 23 days for pairs deployed in 35-40 km. For largest initial separation distances (Figure 2d), the difference between the curves of $4E$ and $\left\langle \delta v^2(t, D_0) \right\rangle$ is approximately constant in time.

Figure 4 focuses on the time evolution of $\left\langle \delta v^2(t, D_0) \right\rangle$ using a logarithmic time scale; in the first day and after the fifth day the trend of $\left\langle \delta v^2(t, D_0) \right\rangle$ is constant for all the initial separation distances; it increases quasi-linearly between 2 and 3 days

for initial distance of 1-3 km.

The temporal evolution of the relative dispersion calculated from Eq. 1 is depicted in Figures 5 and 6 for different initial separation distances. These figures show the existence of 4 regimes: Lundgren, Richardson, Ballistic and Rayleigh. Figure 5 is focused on the time evolution in the first 3 days after deployment and show an exponential grow of relative dispersion. The exponential regime is observed for all the initial distances $D_0$ and evolves as $D^2(t, D_0) = \left\langle D_0^2 \right\rangle e^{\alpha t}$, where $\left\langle D_0^2 \right\rangle$ is the

mean of initial separation distances and $\alpha$ is the growth rate. The values of $\left\langle D_0^2 \right\rangle$ (2.38 km, 37 km, 1200 km and 3000 km) and the range of values of $\alpha$ (2.64, 1.62, 0.64 and 0.35) relate to $D_0$ (initial pair separations of 1-3 km, 5-10 km, 35-40 km and 55-60 km, respectively) can be observed in detail in Figure 5. For $D_0$ smaller than 10 km the relative dispersion is exponential in the first 1.5 days, whereas between 1.5 days and 3 days it follows the Richardson law for $D_0 < 3$ km and the Ballistic law for $D_0 > 5$ km, respectively (Figure 6).

The Richardson law, corresponding to a cubic growth with time, appears for time ranging from 1.5 to 7 days and for small values of $D_0$ (less than 3 km; Figure 6). The dispersion is ballistic between 1.5 days and 13 days and for values of $D_0$ of 5-10 km. The dispersion grows as $D^2(t, D_0) \sim t^\alpha$, with a power $1 < \alpha < 2$, between 5 and 13 days for $D_0$ in 35-40 km. The dispersion is diffusive after 13 days for $D_0$ in 35-40 km and after 3 days for $D_0$ in 55-60 km. In the Rayleigh regime, a linear growth of relative dispersion starting from 34 days after deployment and 23 days after deployment for initial distances of 1-3

km and 35-40 km, respectively (see Figure 2).

The evolution of the relative diffusivity as a function of the initial separation distance for different $D_0$ also shows three phases, related with the Lundgren, Richardson, and Rayleigh regimes of relative dispersion (Figure 7). The exponential phase is an asymptotic law of $D^2$, defined as the enstrophy cascade range, occurring for distances smaller than 11 km and initial separation distances in $5 < D_0 < 10$ km (Figure 7b); whereas for initial separation distances of 35-40 km the exponential phase

occurs for scales smaller than 45 km (Figure 7c). In the Richardson phase, the diffusivity grows approximately as distance to the 4/3 power for the scale ranging 3 to 15 km and for $1 < D_0 < 3$ km, in agreement with an inverse cascade of energy (Figure 7a). For $35 < D_0 < 40$ km and $55 < D_0 < 60$ km, the relative diffusivities are approximately constant during the Richardson phase (Figures 7c and 7d). At large scales (100 km or more), the relative diffusivities are constants for all values of $D_0$ and all particle velocities are uncorrelated.

The relative characteristic dispersion time at small scales is constant (enstrophy cascade range; Babiano et al., 1990; Ollitrault et al., 2005) and corresponds with the exponential phase of the relative dispersion (Figure 8). The characteristic





dispersion time as a function of initial separation distances of drifter pairs, exhibits a power law $D^{\frac{2}{3}}$, for scales ranging between 3 km and 15 km and $1<D_0<3$ km (Figure 8a). This result is a good indication of the Richardson regime in the inverse energy cascade range. For scales of about 100 km or more, the characteristic dispersion time is constant for different $D_0$ and the regime is diffusive.

**4 Discussion and Conclusions**

In the present work, we have studied the statistical properties of relative dispersion using the dataset of drifters deployed in the SWM. We mainly focus on the initial phase of the relative dispersion (first 40 days) and on space scale ranging from 1 km to the mesoscale (55-60 km).

The relative dispersion, diffusivity and characteristic dispersion time of the surface drifter pairs are in accordance with the

theory of two-dimensional turbulence and with the four classical Lundgren, Richardson, Ballistic and Rayleigh regimes. The Lundgren regime (dispersion increases exponentially in time; non-local regime) occurs during the first few days and depends of the initial pair separation: it is observed at spatial scales less of 8-11 km for $5<D_0<10$ km, and between 37 and 45 km for $35<D_0<40$ km (Figure 5). The diffusivity has a power law of $D^2$ and the characteristic dispersion time is constant, corresponding to an enstrophy cascade (Figures 7 and 8). Similar results on this regime in the surface of World Ocean have

been observed using in-situ measurements, e.g. in the Gulf of Mexico (LaCasce and Ohlmann, 2003), in the Santa Barbara Channel (Ohlmann et al., 2012) and in the southwest Pacific sector (Sebille et al., 2015). In the Mediterranean Sea, the exponential growth of relative dispersion spans to 6 hours after deployment in the Gulf of La Spezia (Haza et al., 2010) and 4–7 days in the Liguro-Provençal Sea (Schroeder et al., 2011). In particular in the Western Mediterranean, Nefzi et al. (2014) found the exponential regime at small time, lasts only 3 days using a high-resolution primitive equation model. The existence

of non-local exponential regime reveals a dominant effect of large mesoscale eddies in controlling the transport of surface waters.

The Richardson regime takes place at intermediate time (from 1.5 days to roughly 7 days) and for small $D_0$ ($D_0<3$ km); the diffusivity is a power law $D^{\frac{4}{3}}$ and the characteristic dispersion time is a power law $D^{\frac{2}{3}}$ which are the signature of an inverse energy cascade (Figure 6a). The local Richardson regime occurs from scale ranging between 3 and 11 km, implying

the existence of small mesoscale structures with similar scale.

The Ballistic regime follows the exponential phase for $5<D_0<10$ km, time scale ranging from 1.5 to 13 days and separation distance between 30 and 100 km (Figure 6b); similar results have been observed in the Adriatic Sea from Poulain et al. (2013).

For $D_0$ of 35-40 km, the exponential regime is followed by a 'mixed' regime which is in the middle between the ballistic and

the Rayleigh regimes (Figure 6c).

The Rayleigh regime started at sufficiently long time (after 34 days) for $D_0$ less than 3 km and after 30 days for $5<D_0<10$ km, whereas it follows the exponential phase for $D_0$ larger than 55 km (Figure 6d). For scales of about 100 km or more, the relative diffusivity and the characteristic dispersion time are approximately constant since all pair relative velocities are uncorrelated (Figures 7 and 8).



**Acknowledgement**

We thank all the people who deployed drifters in the Western Mediterranean and kindly shared their data. This work was supported by the Ministry of Higher Education, Scientific Research and Technology of Information and Communication in the frame work of the Tuniso-Italian collaboration between; the Université de Tunis El Manar, Faculté des Sciences de Tunis, Unité du Rayonnement Thermique, Tunisie and National Institute of Oceanography and Experimental Geophysics, Sgonico (Ts), Italy. We are grateful to Antonio Bussani and Thomas Miraglio for heir technical advice about drifters.

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





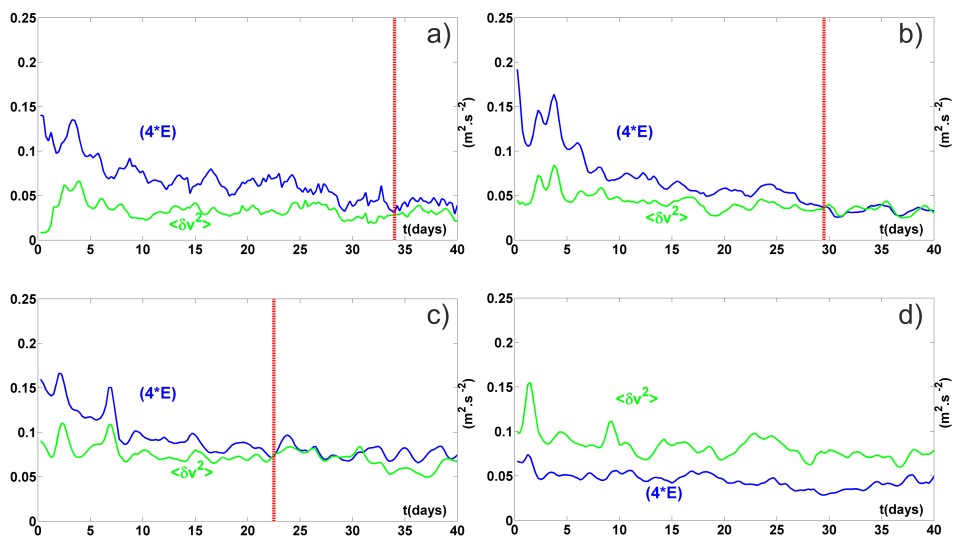

**Figure 2.** Mean square relative velocity $\left\langle \delta v^2 \right\rangle$ and four time kinetic energy 4E for initial pair separations of 1-3 km (a),  5-10 km (b), 35-40 km (c) and 55-60 km (d) in function of time after deployment. The vertical dashed lines show the time at which the pair velocities beco uncorrelated.

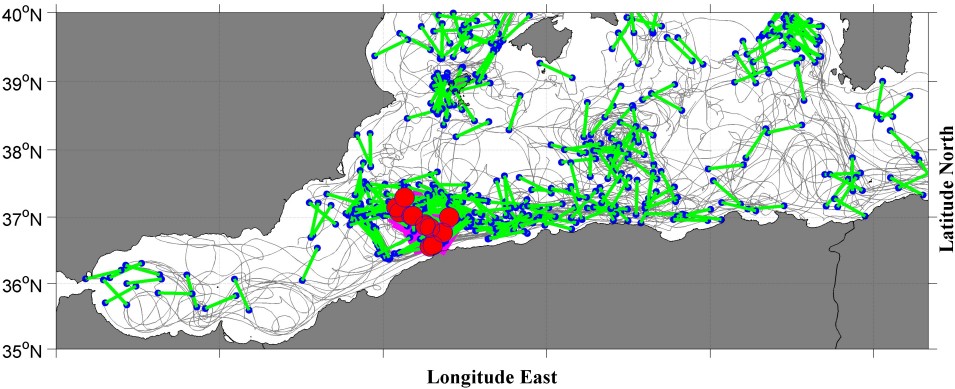

**Figure    3.**

**Trajectories of all pairs (original and chance pairs) and initial positions of the pairs with initial separation distance of 55-60 km. The original pairs (red circles) and chance pairs (blue dots) are connected respectively with magenta and green lines.**



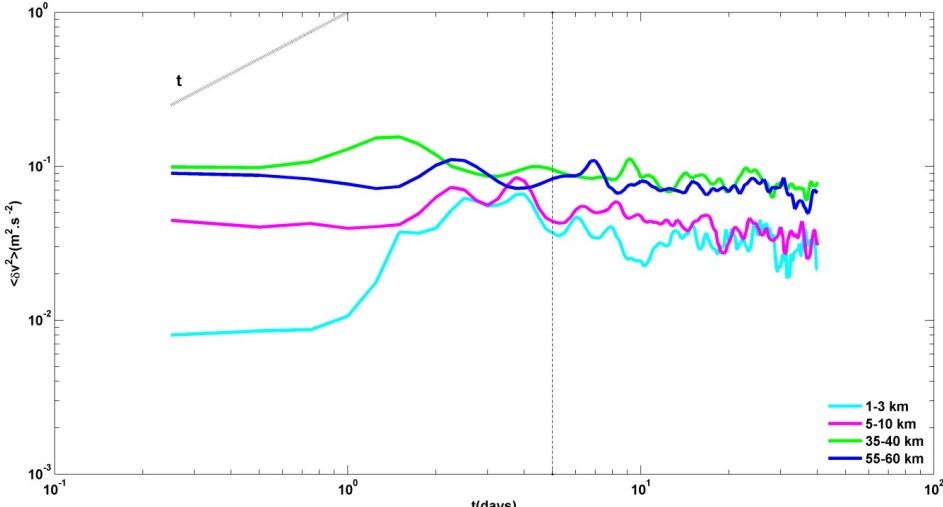

**Figure 4. Mean square relative velocity versus time after deployment (logarithmic scale) for initial pair separations of 1-3 km, 5-10 km, 35-40 km and 55-60 km. The vertical grey line correspond to 5 days.**

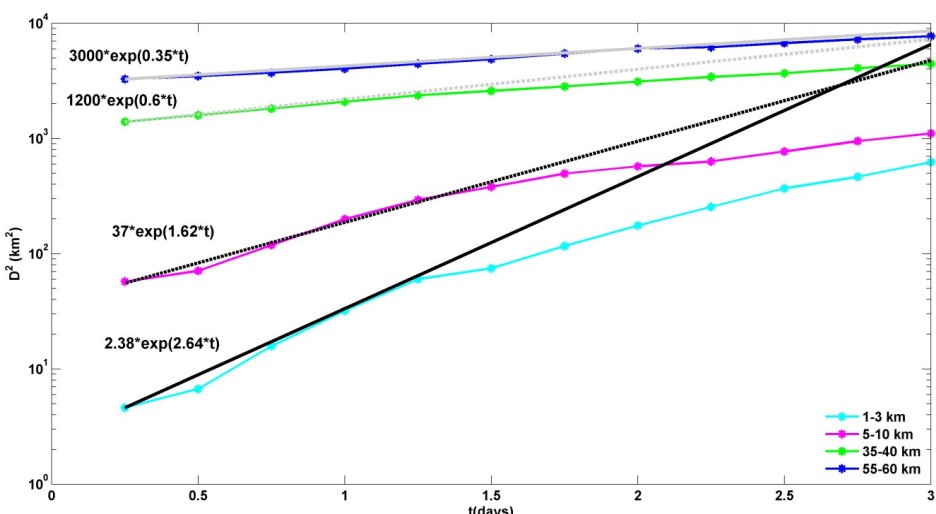

**Figure 5. Relative dispersion in a log-linear plot as a function of time after deployment for initial pair separations of 1-3 km, 5-10 km, 35-40 km and 55-60 km. Straight lines represent the experimental fits.**





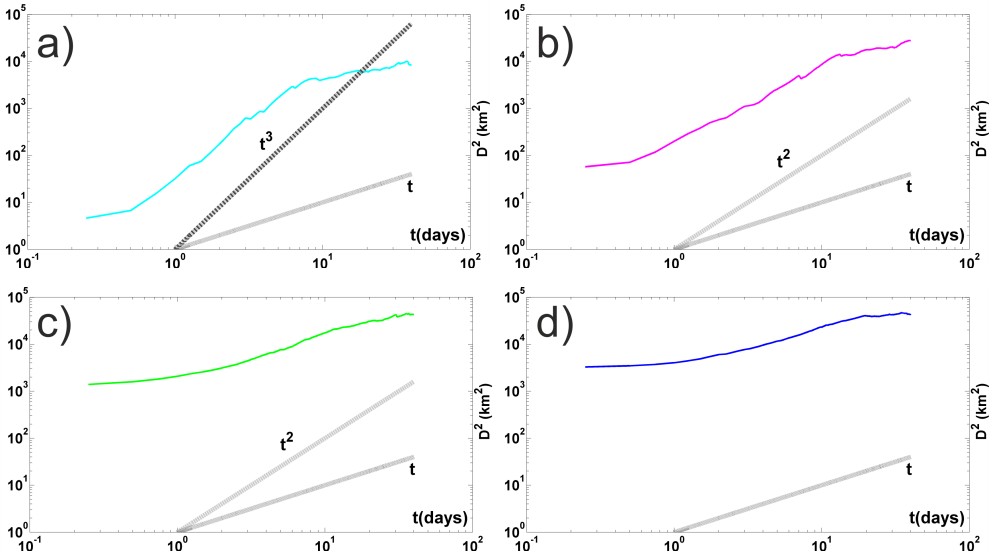

**Figure 6. Relative dispersion versus time after deployment of initial pair separations of 1-3 km (a), 5-10 km (b), 35-40 km (c) and**
5    **55-60 km (d).**

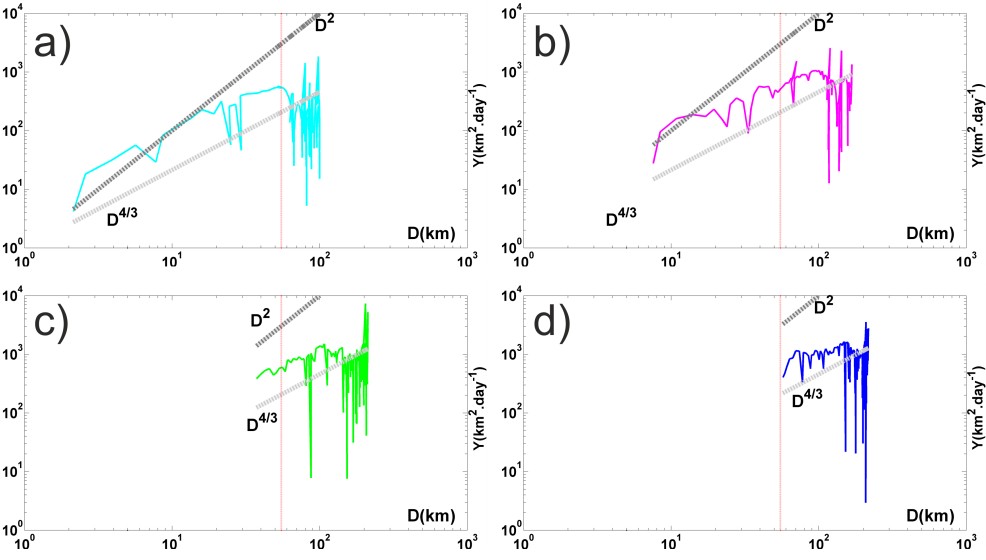

10   **Figure 7. Relative diffusivities in function of separation distance for initial pair separations of 1-3 km (a), 5-10 km (b), 35-40 km (c)**
     **and 55-60 km (d). The vertical red dashed lines show the distance at which the pair velocities become uncorrelated.**





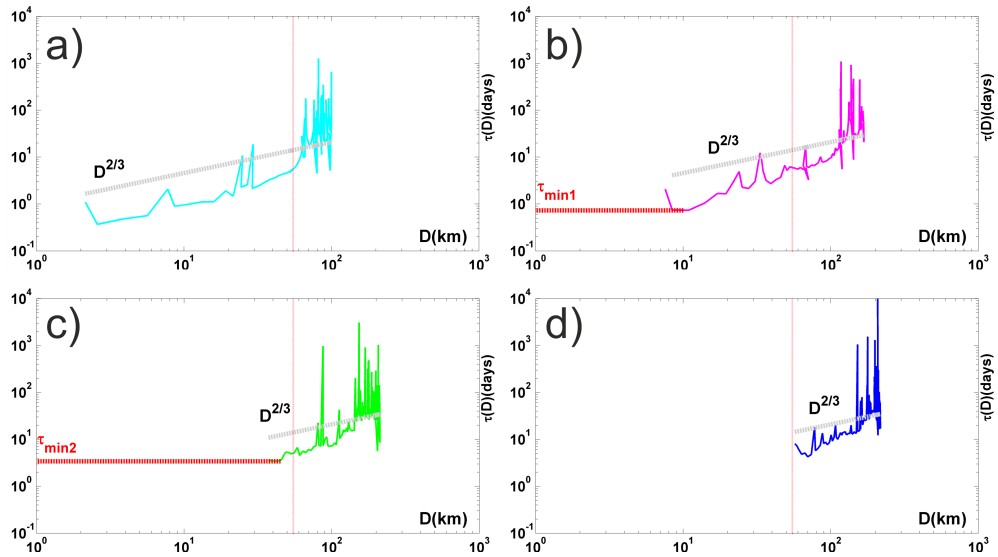

**Figure 8. Characteristic dispersion times in function of separation distance for initial pair separations of 1-3 km (a), 5-10 km (b), 35-40 km (c) and 55-60 km (d). The vertical red dashed lines show the distance at which the pair velocities become uncorrelated.**

