# Peer review of "Relative dispersion in the South Western Mediterranean as derived from satellite-tracked surface drifting buoys"

_Ocean Science, 2017_

## Referee Comment (RC1) · Anonymous Referee #1 · 31 May 2017

Review of the manuscript: "Relative dispersion in the South Western Mediterranean as derived from satellite-tracked surface drifting buoys", by Maher Bouzaiene, Milena Menna, Pierre-Marie Poulain and Dalila Elhmaidi.

General considerations: This work appears rather superficial and fragmentary, without a clear guiding line. The kind of data analysis proposed and the presentation of the results do not respect, in my opinion, the minimal requirements to consider this manuscript for publication. The Authors should decide if – in their opinion – it is better to separate the relative dispersion regimes they claim to observe in the time domain or as function of the separation scale. Physical arguments demand a description of the relative dispersion regimes in function of the separation scale, not in function of time.

This fact has been established, and repeatedly confirmed, by many papers published in the last 20 years about which the Authors seem to be totally unaware. For example, at Page 2, Line 20: "In the last decades several studies have been focused on the relative dispersion in many parts of the World Ocean…." That's true. Unfortunately, the Authors seem to ignore several works that could have been useful to know before undertaking a study like this. For example, I would like to suggest a paper like this one: "General characteristic of relative dispersion in the ocean", Corrado et al., Nature Sci. Rep. 2017, and references therein, where it is explained how to measure the physical characteristics of the relative dispersion process in the correct way, what is the optimal indicator of relative dispersion and how to interpret the results in a global theoretical picture. If this paper may seem too recent to be cited (in the present version of this manuscript), there is for example a very good review about Lagrangian statistics in ocean and atmosphere by LaCasce, "Statistics from Lagrangian observations", Progr. in Oceanogr., 2008, that could have been included in the bibliography. It is not necessary to go into a detailed list of remarks. Just some further considerations. Fig. 2. Diffusion is an asymptotic regime, i.e. it holds in the limit of t $\rightarrow$ infinity, or better for infinitely large separation distance. That's true that the diffusive regime may exist far before these limits are reached, namely, it can occur right after the spatial correlations between trajectories have decayed. Of course, it is simpler to look at single trajectory autocorrelations. In that case, velocity autocorrelation functions could help to see on what time scale the autocorrelations decay. By the way, it is singular that the d) panel shows apparently no correlation decay since particles having a large initial separation should reach the diffusive regime (if any) presumably earlier than pairs starting from much smaller initial separations. Fig. 4. It is supposed that the mean square relative velocity should converge to some constant value for any initial separation, and this is clearly not the case shown in the figure. What is the meaning of the vertical line and how it is related to the vertical lines shown in Fig. 2 ? Fig. 6. The Authors have discovered that the mean square relative dispersion depends significantly on the initial separation, if measured as function of the time… see for example: "Nonasymptotic
properties of transport and mixing", Boffetta et al., Chaos, 2000. That's why some people have introduced and developed the so-called scale-dependent analysis of relative dispersion in terms of the indicator known as Finite-Scale Lyapunov Exponent, or FSLE. The Authors might not recognize or might underestimate the value of this innovation but scientists are supposed to be open to welcome new instrumentation or new analysis techniques that could help them to do their work better than in the past. Fig. 7. Where the vertical lines come from, and for what reason pair velocities should be uncorrelated during a Richardson's regime ? Velocities become uncorrelated only in the standard diffusion regime. Before diffusion, any other pre-asymptotic regime is characterized by the persistence of spatial and temporal correlations. When all correlations decay, relative dispersion tends to a standard diffusive regime (constant diffusivity). Fig. 8. Again, if the vertical lines actually indicate the separation after which the pair velocities become uncorrelated (let us assume this is true), I don't understand why the characteristic dispersion time doesn't seem to scale as D2 (as in a diffusive regime), first of all. Second, it does not make sense to look for a D2/3 fit (as in a Richardson scaling) at scales larger than the presumed spatial correlation length (panels b, c and d), since in a Richardson turbulent dispersion regime particle pairs are still correlated by the presence of eddies at the same scale of the particle separation (locality property).

I cannot recommend publication of this manuscript. I invite the Authors to pay much more attention to the writing of the text and, above all, to the presentation and the discussion of the results in more rigorous terms. Some fundamental literature is missing from the reference list and this explains, in part, why a study on Lagrangian relative dispersion of ocean drifters, submitted in 2017, looks so out of date and surprisingly superficial.
* * *

---

## Referee Comment (RC2) · Anonymous Referee #2 · 20 Jun 2017

In this manuscript, the authors analyse a set of surface drifters in the western Mediterranean for dispersion characteristics. They focus on time scales up to a few weeks and spatial scales up to a few tens of kilometers, and find agreement with previous studies in other parts of the ocean, as well as the Mediterranean itself.

While I am happy to see more of these types of studies into the surface dispersion being done, I found this one a rather disappointing read. The manuscript does not really add anything new to the vast corpus of publications on the topic of pair-wise surface drifter dispersion (as the authors also seem to acknowledge when they compare their results to many previous ones).

[Figure]

I think that this study is a missed opportunity to go much deeper and discover new things we didn't yet know about the ocean. Therefore, I'm afraid that unless the authors come back with something novel, I will have to recommend rejection of this manuscript to Ocean Science on the basis of lack of novelty.

Beyond that insurmountable point, I have the following major issues:

- There is no discussion of confidence levels and statistics. Are any of the fits statistically robust? What is the probability that we are not simply looking at noise?

- All drifters are lumped together, and this is a missed opportunity. Is there any variation in dispersion characteristics between drifter type, region within the Western Mediterranean, time period, season or anything else?

- Why did the authors decide to analyse the data in terms of dispersion versus time? Many other studies use Finite Scale Lyapunov Exponents. What would the results be in that framework?

- The motivation for this study in the introduction and conclusion sections can be greatly expanded. Why is this region important? Why would it be different or the same as other oceanic regions? Why would one even care about dispersion regimes?

Then, I also have some minor comments:

- The authors somehow failed to reference the big and seminal review paper by La-Casce (2008) on surface drifter dispersion in Progress of Oceanography

- Figure 4-8 could probably be combined in one figure, that shows the dispersion as a function of time for the entire time series. This would leave space for other, more profound analysis

———————————————

---

## Referee Comment (RC3) · Anonymous Referee #3 · 23 Jun 2017

The manuscript describes the relative dispersion of surface drifter pairs in the South Western Mediterranean (SWM) Sea. The topic has a high scientific relevance because the authors attempt to quantify the surface Lagrangian dispersion at the entrance of the SWM, where Atlantic water enters the region. However, the scientific quality of the study is rather modest. The authors invoke classical dispersion regimes, but these are not sufficiently discussed, and perhaps not very well understood. The arguments and plots to support the presence of different regimes in the SWM are not convincing. In terms of the presentation quality, the manuscript requires major revisions: several sentences seem incomplete or they are very obscure; dispersion plots do not show or mention confidence intervals, statistical significance nor any other argument to support

their robustness. This should be done in a revised version.

Some specific comments are the following. There are no clear interpretations or discussions in some cases, and I will mention only three examples. First, the authors argue that the ballistic behavior is observed in other regions, e.g. in the Adriatic Sea and in the Santa Barbara Channel. However, the length and time scales in the Santa Barbara Channel are completely different, and in that case it is argued that shear dispersion might play a role on the shape of the dispersion curve (Ohlmann et al. 2012). What is the physical mechanism responsible for the apparent ballistic regime in the SWM? To examine this question, I think the authors should take into account more carefully the role of relevant circulation features in the region, namely, the strength, size and direction of the eastward Algerian Current (see Salas et al, J. Mar. Sys. 2001). A second point that demands further discussion is the behavior of the relative velocity in Figures 2 and 4. Panels a, b and c suggest that the velocities between particles are decorrelated after 25-35 days. It sounds that this is a long time, but the authors do not argue very much on why. Is it related with the presence of very persistent currents? A similar result would be found if only drifters near the North African coast are taken into account? Furthermore, the squared relative velocity is often discussed as a function of the pair separations (see e.g. Beron-Vera and LaCasce, 2016) in order to identify the decorrelation length scales; then, further inferences can be made regarding 2D turbulence or shear dispersion regimes. A third example is the interpretation that the Lundgren regime is observed for large initial separations. Other studies have indeed observed a similar exponential growth, but the reason cannot be attributed to the exponential regime in 2D turbulence, which relies on small initial separations within the enstrophy cascade subrange. So, if the authors report exponential growth is fine, even when the particles are initially very far from each other, but the interpretation should be more careful.

One way to improve the study (besides including statistical significance tests) is by exploring more metrics, and not relying only in the time-dependent relative dispersion.

[Figure]

The authors should try to look at higher statistical moments or, more importantly, to the PDFs of separations. There are also scale-dependent metrics that might be considered (FSLEs). My general comment is that the statistical analyses of surface drifters might be incomplete because the observational evidence is scarce; thus, the actions to fix this problem is to consider a more ample number of statistical metrics. Also, it seems important to distinguish whether drifters in some particular regions generate some bias in the statistics, which reflects the relevance of local circulations.

One more comment: Two types of surface drifters with different nominal depths are considered, CODE (1 m) and SVP (15 m). The text should describe how many drifters of each type were considered, because in principle their functioning is different. The CODE drifters might be strongly influenced by the wind, while SVP drifters with a drogue tend to follow geostrophic currents, depending on the surface Ekman layer thickness. There should be some justificatory arguments to use both types in the statistics. Or, ideally, statistics calculated separately should be similar.

---

## Referee Comment (RC4) · Anonymous Referee #4 · 1 Jul 2017

This manuscript presents simple analyses of basic two point statistics from a modest ensemble of surface drifters released in the South Western Mediterranean over the past 20 years. The authors claim that the results confirm, or are at least consistent with four standard scaling law regimes.

While the effort to synthesize existing surface drifter data in this region is certainly welcome, as is the attempt to organize and catagorize the results in the context of previously observed or predicted scaling behavior, the overall scientific quality of the present manuscript falls short of that required for publication. Most importantly, there is a distinct lack of quantitative analysis of the results to support the broad claims made

in the abstract. Instead, the existence of various scaling regimes is vaguely intimated by simply plotting lines of different slope on log-log plots (often with axes extended well beyond the bounds of the data). Given the increasing availability of surface drifter data and the potential ability to connect two-point Lagrangian statistics to spectral properties of fluctuating surface velocity field, understanding and quantifying the existence of scaling regimes, their dependence on initial separation distance and how such properties vary between geographic locations is an important problem. Unfortunately, it is not at all clear how the results as currently presented in this manuscript clarify any existing questions about surface dispersion.

Specific points: (1) The authors concentrate on relative dispersion and its time derivative without providing any measure of the variability in these quantities or the statistical significance of the plotted estimates. What are reasonable error bars on the plots? How well (or poorly) is the data fit by any of the proposed scaling laws? Some reasonable, standard statistical hypothesis testing should be done to support the authors' claims.

(2) The choice of metrics is a bit narrow, and somewhat redundant. Diffusivity and 'characteristic dispersion time' are derivative measures of the relative dispersion, itself simply the time-dependent second moment of the distribution of separation distances. Given that the second moment is dominated large separation pairs, Richardson himself argued for consideration of the pdf. The finite-scale Lyapunov exponent (FSLE) would be a natural metric to consider when searching for exponential regimes.

(3) The authors combine data from two types of drifters, sampling different depths and presumably possessing different response properties. Can one show that these two instruments sample similar velocity fields? Are the resulting distributions of two-point separation statistics statistically similar?

(4) What is the effect of the substantial amount of data smoothing on the results? Is this the same for both the very near surface and deeper drifters? As explained in

section 2, the data are low-pass filtered with a 36 hour cutoff, 'to remove tides and inertial oscillations'. The effects of this smoothing are clearly seen in the results at early times. One could imagine that this smoothing preferentially effects the behavior at smaller initial separation distances. What is the reader to make of the short-time, 1-2 day results in the presence of such averaging?
* * *

---

## Author Comment (AC1) · 24 Jul 2017

Dear all,

We thank the four anonymous Reviewers for their constructive and incisive remarks. As the reviewers themselves have pointed out, this paper needs substantial improvements, several corrections and a more detailed statistical analysis. For this reason, we have decided to retrieve it from OS, waiting to prepare a more suitable work according to the reviewers' suggestions.

Thank you for your time and commitment. With my best regards,

[Figure]

Milena Menna